# Prior Guided 3D Medical Image Landmark Localization

**Yijie Pang**[1]                             12232142@MAIL.SUSTECH.EDU.CN
**Pujin Cheng**[1]                            12032946@MAIL.SUSTECH.EDU.CN
**Junyan Lyu**[1,2]                                JUNYAN.LYU@UQ.EDU.AU
**Fan Lin**[*3]                                    FOXETFOXET@GMAIL.COM
**Xiaoying Tang**[*1]                          TANGXY@SUSTECH.EDU.CN

[1] *Department of Electronic and Electrical Engineering, Southern University of Science and Technology, Shenzhen, China*

[2] *Queensland Brain Institute, The University of Queensland, St Lucia, QLD, Australia*

[3] *Department of Radiology, Shenzhen Second People's Hospital, Shenzhen, China*

**Editors:** Accepted for publication at MIDL 2023

## Abstract

Accurate detection of 3D medical landmarks is critical for evaluating and characterizing anatomical features and performing preoperative planning. However, detecting 3D landmarks can be challenging due to the local structural homogeneity of medical images. In this study, we present a prior guided coarse-to-fine framework for efficient and accurate 3D medical landmark detection. Specifically, we utilize the prior knowledge that in specific settings, physicians often annotate multiple landmarks on a same slice. In the coarse stage, we perform coordinate regression on downsampled 3D images to maintain the structural relationships across different landmarks. In the fine stage, we categorize landmarks as independent and correlated landmarks based on their annotation prior. For each independent landmark, we train a single localization model to capture local features and deliver reliable local predictions. For correlated landmarks, we mimic the manual annotation process and propose a correlated landmark detection model that fuses information from various patches to query key slices and identify correlated landmarks. The proposed method is extensively evaluated on two datasets, exhibiting superior performance with an average detection error of respective 3.29 mm and 2.13 mm.

**Keywords:** 3D medical landmark detection, coarse-to-fine, prior knowledge.

## 1. Introduction

Anatomical landmark localization is crucial for various types of medical applications, including orthodontic and maxillofacial surgery planning (Wang et al., 2016), organ volume estimation (Turkbey et al., 2019), and skeletal development assessment (Escobar et al., 2019). Since manual annotation of anatomical landmarks is cumbersome and labor-intensive, accurate and automatic landmark detection is highly desired. Unfortunately, medical images' local structural similarity often leads to ambiguity in landmark localization, especially in 3D cases wherein adjacent slices have similar structural and intensity profiles. In such context, physicians often annotate landmarks with specific landmarks serving as references (Wang et al., 2016) or annotate multiple landmarks on a same key slice (Turkbey et al., 2019; Tan et al., 2022). As shown in panel (a) of Figure 1, the occipital bone landmark can be

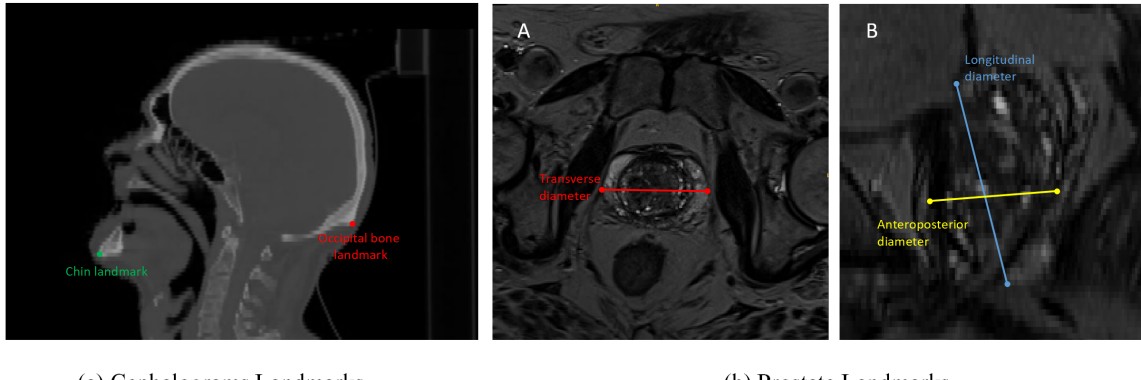

(a) Cephalograms Landmarks          (b) Prostate Landmarks

Figure 1: Representative examples of correlated landmarks.

localized by referring to the chin landmark. In panel (b) of Figure 1, multiple landmarks sit on the same slices having the largest organ areas. In these cases, solely focusing on the local features of an individual landmark may misidentify the key slice and induce large localization deviations. Accurately identifying those key slices shall well improve the accuracy of localizing correlated landmarks.

With the emergence of 3D convolutional neural networks (CNNs) (Çiçek et al., 2016), 3D landmark detection methods have advanced from key slice classification (Yang et al., 2015) to coordinate and heatmap regression with 3D CNNs. However, processing 3D volume data is computationally expensive due to the increased network parameters. Typically, 3D medical images are downsampled to of relatively low resolution for regression-based landmark detection. Zhang et al. establish a multitask model for image volume classification and landmark localization (Zhang et al., 2020a). Liu et al. simultaneously predict landmarks' positions and the inter-landmark distances for THA preoperative planning (Liu et al., 2020). Payer et al. introduce a method that combines local appearance with spatial configuration for heatmap regression and achieves excellent results on multiple datasets (Payer et al., 2019). These approaches need to first downsample the entire image, which weakens local texture feature extraction and may cause inevitable errors. Numerous studies demonstrate that integrating high-resolution information is beneficial for landmark localization (Zhang et al., 2020b).

Coarse-to-fine strategies may relieve the accuracy degradation issue caused by downsampling. Tao and Zheng employ a transformer-based structure for coarse landmark detection, followed by a Unet model for fine landmark detection (Tao and Zheng, 2021). Zeng et al. propose a three-stage coordinate regression framework that progressively utilizes global and local attention (Zeng et al., 2021). Lee et al. integrate global and local feature extractions as well as specific shape constraints into a single-passing CNN for landmark localization (Lee et al., 2022). Chen et al. propose a Structure-Aware Long Short Term Memory (SA-LSTM) framework that fuses the information from multi-resolution patches to continuously adjust the landmark offsets (Chen et al., 2022). Most of these methods employ coordinate regression in the fine stage, which has an innate potential to incorporate implicit structural knowledge (Jin et al., 2021). However, the global average pooling operation employed in

these methods may undermine the spatial structure and impose a detrimental effect on precise landmark localization (Mao et al., 2022).

In addition to the aforementioned methods, deep reinforcement learning (RL) has also been employed for localizing various anatomical landmarks utilizing patch-based information. Alansary et al. employ an artificial RL agent to learn the optimal path for landmark localization using multi-scale searching strategies (Alansary et al., 2019). Bekkoucha et al. introduce a multi-agent RL approach that combines graphical lasso and Morris sensitivity analysis to accurately quantify the impact of specific landmark subgroups on the localization of other landmarks (Bekkouch et al., 2022). Search-based Activate Appearance Model (Gao et al., 2010) can also model the prior knowledge of landmarks' shape and texture. However, compared with heatmap-based methods, search-based ones tend to be more time-consuming at the inference phase.

Although most of the existing methods implicitly integrate structural knowledge (Gao et al., 2010; Jin et al., 2021; Zeng et al., 2021), the integrated knowledge is kind of weak, since it is usually indirectly inferred from the landmarks' distributions. Moreover, these methods tend to ignore the physicians' prior knowledge when manually annotating specific landmarks (e.g. correlated landmarks). Physicians' prior knowledge indicates the relevance of different landmarks and shall be more explicit and more useful. Incorporating physicians' prior knowledge into the automated landmark localization process is crucial, which nevertheless is a relatively unexplored research area.

We here propose a prior guided coarse-to-fine landmark localization framework to effectively combine the advantages of heatmap regression and coordinate regression and integrate the prior knowledge from physicians' annotation process. We adopt coordinate regression in the coarse stage to preserve structural constraints across landmarks (Jin et al., 2021). In the fine stage, we categorize the landmarks into independent and correlated ones based on the physicians' annotation process. For independent landmarks, we train multiple Unet models to fully focus on the local features of each landmark. For correlated landmarks located at the same slices, we propose an axial attention fusion module and a key slice detection module. These two modules can fuse correlated patches' features to attain a high key slice detection accuracy and assist in detecting ambiguous landmarks.

Our contributions can be summarized as follows: (1) Combining the advantages of the two types of regression methods as well as physicians' prior knowledge, we propose a novel coarse-to-fine 3D medical image landmark localization framework. (2) To the best of our knowledge, we are the first here to design a correlated landmark localization module guided by prior knowledge. (3) We evaluate our proposed method on two datasets and demonstrate its superiority through extensive experiments. The source code is publicly at https://github.com/pang-yi-jie.

## 2. Methods

Our method employs a coarse-to-fine framework for 3D landmark detection, as illustrated in Figure 2. In the coarse stage, a ResNet34 model is trained for coordinate regression with downsampled images as the input. Subsequently, our fine stage employs patch-based Unet models to perform heatmap regression towards each independent landmark, with cropped regions of interest (ROIs) from the original high-resolution image as the input. For

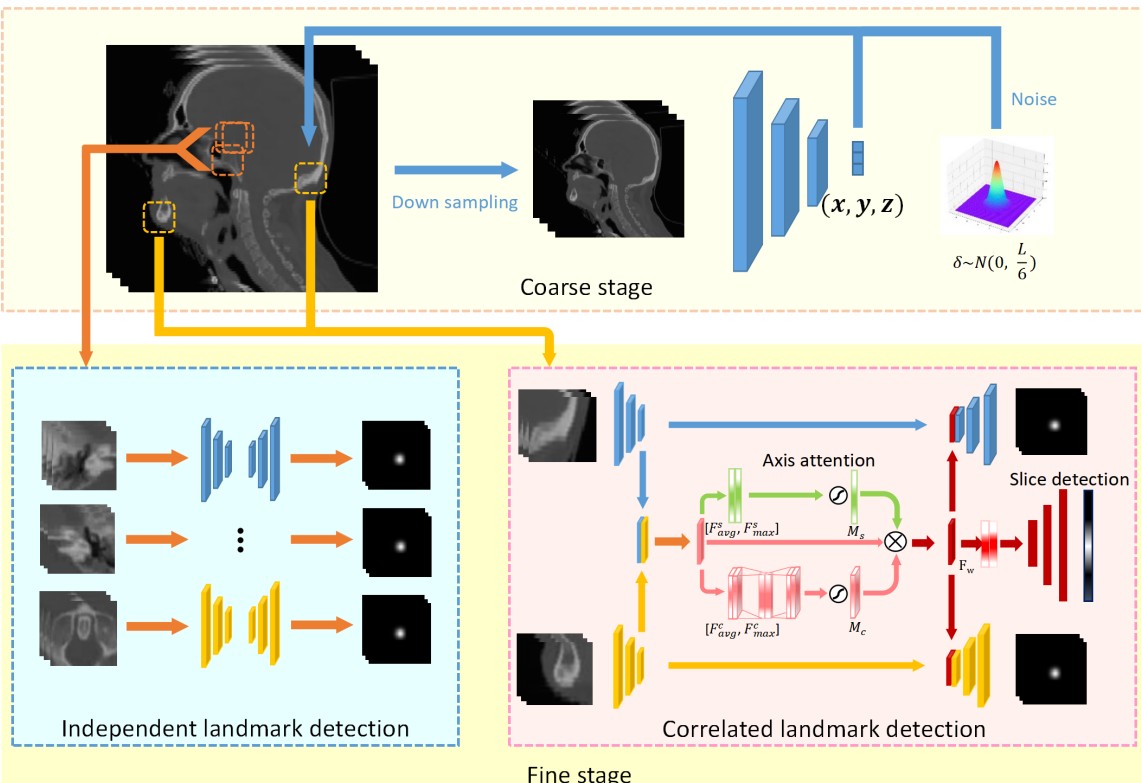

Figure 2: The proposed prior guided coarse-to-fine landmark localization pipeline.

correlated landmarks, we adopt an axis attention fusion module with dynamically weighted refining, along with a key slice detection module to leverage physicians' annotation prior.

## 2.1. Coarse stage

The coarse stage takes the downsampled image as the input. Given $n$ landmarks of interest with ground truth $(\hat{x}, \hat{y}, \hat{z})$, we employ a ResNet-34 (He et al., 2016) for coordinate regression. We modify the output length of the fully connected layer to $3 \times N$ and use the following regression loss to train the network:

$$L_{reg} = \frac{1}{N} \sum_{i=0}^{N} (|x_i - \hat{x}_i| + |y_i - \hat{y}_i| + |z_i - \hat{z}_i|). \tag{1}$$

where $(x, y, z)$ denotes the coarse estimation of the landmarks' positions. Considering the coarse stage's result $(x, y, z)$ shall converge to stable predictions, cropping patches centering on $(x, y, z)$ may result in training samples that are lacking of diversity. Inspired by (Chen et al., 2022), we add Gaussian noise $\delta \sim N(0, \sigma)$ to the cropping centers. We set $\sigma$ to $L/6$ to ensure the resultant patches contain the to-be-predicted landmarks, where $L$ is the patch's edge length. Subsequently, we crop the original high-resolution images based on the modified landmarks' positions and obtain ROI images for fine localization.

## 2.2. Fine stage

The fine stage focuses on extracting local features around multiple landmarks using patch-based Unets (Çiçek et al., 2016). To effectively exploit physicians' prior knowledge, we categorize the landmarks of interest into independent and correlated categories according to the physicians' annotation practice. For independent landmarks, physicians separately annotate them based on local texture features. As shown at the bottom left panel of Figure 2, each Unet extracts features and predicts heatmap for the corresponding patch. For correlated landmarks, physicians typically identify specific slices that contain the entire organ's characteristics and annotate correlated landmarks on those slices. In light of this, we reset the coarsely predicted centers of the correlated ROI patches to be on the same slice. This ensures that the key slice is the same for correlated patches. We design an axis attention module and a key slice detection module for key slice querying and landmark detection.

**Axis attention**     Since we utilize different encoders to handle the correlated patches separately, we deeply fuse axis features for key slice detection. Inspired by the CBAM module (Woo et al., 2018), we propose axis channel attention and spatial attention to perform dynamically weighted refining. Axis is defined as the direction perpendicular to the key slice.

We sequentially concatenate features of the correlated patches and feed them into a 3 $\times$ 3 convolution layer, obtaining a fused feature map $F \in R^{C \times H \times W \times D}$, where $(C, H, W, D)$ respectively represent the channel's number, height, width and depth of the fused feature map. We assume the axis follows the depth direction, so as to build the axis attention module. To encourage the channel attention block to capture spatial interactions from axis information, we perform axis's average pooling through global average pooling over $H$ and $W$. It can preserve axis positional information and generate axis's average channel context descriptor $F_{avg}^c \in R^{C \times D}$. Axis's global max pooling is also conducted to obtain axis's maximum channel descriptor $F_{max}^c \in R^{C \times D}$. Both descriptors are then forwarded to a shared $1 \times 1 \times 1$ convolution layer and fused by element-wise summation. After sigmoid activation $\sigma$, we obtain axis channel attention maps $M_c \in R^{C \times D}$,

$$M_c = \sigma(Conv_{1 \times 1 \times 1}(F_{max}^c) + Conv_{1 \times 1 \times 1}(F_{avg}^c)). \tag{2}$$

We use average pooling and maximum pooling to squeeze the channel information of $[F_{avg}^c, F_{max}^c]$, yielding a pair of direction-aware axis spatial attention maps: $[F_{avg}^s, F_{max}^s] \in R^{2 \times D}$. Feeding it into a convolution layer and sigmoid activation $\sigma$, we obtain the axis's spatial attention map $M_s \in R^D$. After element-wise multiplication of $F$, $M_s$ and $M_c$, we get the dynamically weighted feature maps $F_w = F \times M_s \times M_c$,

$$M_s = \sigma(Conv_{1 \times 1 \times 1}[F_{max}^s, F_{avg}^s]). \tag{3}$$

**Heatmap regression and key slice detection**     Afterward, we concatenate the dynamically weighted features with the patch features to guide patch-based heatmap regression. To improve the localization accuracy in the axial direction, we introduce a slice detection branch to determine the key slice and constrain the attention map. Same as axis channel attention, we use pooling operations to preserve precise positional features along

the axis direction and use 1D convolutions to decode the high-level features. Unlike traditional classification, we use a Gaussian heatmap to represent the probability of the key slice's location. The loss functions for slice detection $L_{slice}$ and patch heatmap regression $L_{patch}$ are both formulated with MSE, and the overall loss function for the fine stage is:

$$L_{fine} = \lambda_0 L_{patch} + \lambda_1 L_{slice}, \tag{4}$$

where $\lambda_0$ and $\lambda_1$ are the weights of patch heatmap regression and slice classification. In the inference phase, the DARK (Zhang et al., 2020b) method is adopted to decode the two different heatmaps, obtaining the locations of landmarks and key slices.

## 3. Experiments and Results

### 3.1. Datasets

**In-house prostate dataset** The in-house prostate dataset consists of 857 T2-weighted MRI scans with a slice space of 0.67 mm, acquired from the Health Science Center, Shenzhen Second People's Hospital. The scans are saved in NII format and contain six prostate landmarks annotated by an experienced clinician for prostate volume estimation (Turkbey et al., 2019). The six landmarks are the endpoints of the maximum longitudinal diameter, the maximum transverse diameter as well as the maximum anteroposterior diameter. The dataset are randomly split into 514 scans for training, 171 for validation, and 172 for testing.

**PDDCA dataset** The PDDCA dataset comprises 48 CT images selected from the Radiation Therapy Oncology Group 0522 study. The images are in NRRD format with various resolutions and spacings. The dataset were used in the MICCAI Head-Neck Challenge 2015 (Raudaschl et al., 2017), and 33 images were labeled with five bony landmarks for gross alignment evaluation. The landmarks include the right condyloid process landmark (mand_r), the left condyloid process landmark (mand_l), the chin landmark (chin), the odontoid process landmark (odont_p), and the occipital bone landmark (occ). Following a previous work (Chen et al., 2022), we resize all images to 518×518×384 with an isotropic voxel spacing of 1 mm. The dataset are equally divided for three-fold cross-validation.

### 3.2. Implementation details

Our method is implemented using PyTorch and trained on an NVIDIA GeForce RTX 2080 Ti GPU. The learning rates for the coarse and fine stages are respectively set to be $1e^{-3}$ and $1e^{-6}$. The training epochs are 50 on the prostate dataset and 300 on the PDDCA dataset. We use the SGD optimizer with a weight decay of $1e^{-8}$. Random translation (10% of the input size), random scaling (up to 10% difference), and random intensity shift (up to 0.1) are applied for data augmentation. The input size of the coarse stage is set to be 64×64×64 for the prostate dataset and 96×96×72 for PDDCA. To balance the trade-off between computational memory consumption and contextual feature extraction, we set the cropping size to be 48×48×48 in the fine stage and the cropping resolutions to be respective 1 mm and 2 mm for the two datasets.

### 3.3. Comparison to state-of-the-art

To evaluate the performance of our proposed method, we conduct comprehensive experiments on the two datasets. We compare our method to three baseline deep learning-based medical landmark localization methods: 3D Unet (Çiçek et al., 2016), SCN (Payer et al., 2019), and DRM (Zhong et al., 2019). All methods are assessed with the evaluation metrics proposed by (Chen et al., 2022), including the mean radial error (MRE), standard deviation (SD), and successful detection rate (SDR) at five target radii (2 mm, 2.5 mm, 3 mm, 4 mm, 8 mm). For one-stage methods, all 3D images are downsampled to the maximum size allowed by memory as the input. Two-stage methods use the same hyperparameters as our proposed method, such as the downsampling volume, patch size, and spacing.

Table 1: Performance comparisons on the two datasets. † and ‡ respectively denote one-stage methods and coarse-to-fine methods.

| Dataset | Method | MRE (SD) | SDR (%) | | | | |
|---------|--------|----------|---------|---------|--------|--------|--------|
| | | | 2 mm | 2.5 mm | 3 mm | 4 mm | 8 mm |
| PDDCA | 3D-Unet † | 7.69 (5.24) | 2.03 | 3.25 | 5.05 | 16.28 | 67.90 |
| | SCN † | 7.44 (4.26) | 2.65 | 6.74 | 10.98 | 21.36 | 69.30 |
| | DRM ‡ | 6.39 (3.37) | 7.27 | 12.72 | 16.36 | 29.09 | 74.54 |
| | LA-GCN ‡ | 3.23 (2.52) | 35.68 | 46.76 | 58.19 | 69.48 | 94.74 |
| | SA-LSTM ‡ | 2.37 (1.60) | **56.36** | **71.60** | 80.00 | 89.99 | 95.91 |
| | **Proposed** ‡ | **2.13 (1.18)** | 55.23 | 70.12 | **86.20** | **93.50** | **99.40** |
| Prostate | 3D-Unet † | 3.57 (2.27) | 23.84 | 36.82 | 48.23 | 68.12 | 96.32 |
| | SCN † | 3.48 (2.31) | 25.68 | 39.34 | 51.74 | 69.57 | 95.73 |
| | DRM ‡ | 3.44 (**2.21**) | 26.74 | 38.24 | 52.13 | 70.54 | **97.58** |
| | **Proposed** ‡ | **3.29** (2.26) | **31.17** | **41.67** | **54.13** | **73.22** | 95.62 |

**Results on the PDDCA dataset**   We show our method's results as well as those directly copied from previous literature (Chen et al., 2022) in Table 1. The results demonstrate that two-stage methods generally deliver higher localization accuracies since they can effectively utilize high-resolution features. Among all methods, our proposed method performs the best in terms of MRE (2.13 mm) and SD (1.18 mm). In terms of SDR, our method achieves much higher successful detection rates than the second best-performing method SA-LSTM for the target radii of 3 mm, 4 mm, and 8 mm. For the other two target radii, namely 2 mm and 2.5 mm, the proposed method's performance is on par with SA-LSTM. Detailed analysis of different landmarks' localization performance is presented in Appendix A.1. From those detailed analysis results, the proposed method can more accurately locate ambiguous landmarks, probably because of our specifically-designed axis attention fusion module.

**Results on the in-house prostate dataset**   LA-GCN adopts Mask-RCNN in the fine stage, which makes use of the segmentation labels of bones. As such, LA-GCN does not work for our in-house prostate dataset. Due to the complexity of the SA-LSTM method and the instability of landmarks' local features, SA-LSTM fails to continuously adjust the

landmark offsets on the prostate dataset. Therefore, we only present the comparison results of the three baselines, namely 3D Unet, SCN, and DRM in Table 1. Due to the small volume of the prostate data, one-stage methods can also use high-resolution data as the input and thus achieve decent accuracy. With that being said, our proposed method still obtains the best localization performance, in terms of almost all evaluation metrics. This may be because our method has a stronger local feature extraction capability. More detailed experimental results are presented in Appendix A.2.

### 3.4. Ablation study

The main contribution of our method is the correlated landmark regression module. We conducted ablation experiments on its three components to verify their importance. The fusion module refers to the convolution layer and the information interaction between correlated patches. Without the fusion module, all correlated landmarks will be treated as independent landmarks. The axis attention module and the slice detection branch are the two modules we mainly introduced above. The ablation analysis results on the PDDCA dataset (22 samples for training and 11 samples for validation) are listed in Table 2. The feature fusion module can guide ambiguous landmarks' localization, significantly enhancing performance. The axial attention module and the slice detection branch aggregate axis features to further guide the localization of ambiguous landmarks. These modules enable our model to achieve a low MRE (1.82 mm) on correlated landmarks. Collectively, our proposed method can effectively use prior information from physicians' annotation practice and constrain the locations of correlated landmarks (Appendix A.3).

Table 2: Ablation study results on the PDDCA dataset as evaluated by MRE (SD).

| Component | | | Correlated landmarks | | Independent landmarks | | |
|---|---|---|---|---|---|---|---|
| Fusion | Axis attention | Slice detection | Chin | Occ | Mand_l | Mand_r | Odont_p |
| ✓ | | | 2.00 (0.65) | 4.32 (3.10) | | | |
| ✓ | ✓ | | 1.87 (**0.57**) | 2.18 (1.18) | **2.05 (0.53)** | **2.18 (0.74)** | **1.68 (1.14)** |
| ✓ | ✓ | | **1.76** (0.66) | 2.28 (1.56) | | | |
| ✓ | ✓ | ✓ | 1.85 (0.62) | **1.79 (1.15)** | | | |

### 4. Conclusion

This paper proposes a coarse-to-fine framework for localizing independent and correlated anatomical landmarks from 3D medical images. In the fine stage, we employ multiple Unet models for heatmap regression for independent landmarks, ensuring that each model solely focuses on a patch centering the specific single landmark of interest. For the correlated landmarks, we propose a feature fusion module and a key slice detection module. It successfully identifies the position of the key slice from multiple patches and uses the fused features to assist in landmark localization. Our method outperforms state-of-the-art methods, according to extensive experiments on the publicly-accessible PDDCA dataset and our in-house prostate dataset. We shall further incorporate more types of prior knowledge, such as the shape prior of the anatomical landmarks distribution and other prior knowledge utilized in the manual annotation process in our future work.

## Acknowledgments

This study was supported by the National Natural Science Foundation of China (62071210); the Shenzhen Science and Technology Program (RCYX20210609103056042); the Shenzhen Science and Technology Innovation Committee (KCXFZ2020122117340001); the Shenzhen Basic Research Program (JCYJ20200925153847004, JCYJ20190809120205578); the Special Funds for the Cultivation of Guangdong College Students' Scientific and Technological Innoation (pdjh2023c11018).

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

# Appendix A. More Experimental Results

## A.1 More results on the PDDCA dataset

To convincingly demonstrate the superiority of our approach, we re-implement four deep learning-based methods for extensive comparisons, ie 3D Unet, SCN, DRM, and SA-LSTM. We employ the optimal hyperparameters to maximize their performance, such as the cropping size, kernel size of the Gaussian heatmap and input resolution. The PDDCA dataset are equally distributed into three folds to perform three-fold cross-validation, and the average individual landmark localization accuracy is reported in Table A.1. We also show the SDR curve of our method as well as that of SA-LSTM at different radii in Figure A.1

Table A.1: Individual landmark localization performance comparisons on PDDCA, in terms of MRE.

| Method | Overall | Chin | Mand_l | Mand_r | Odont_proc | Occ_bone |
|---|---|---|---|---|---|---|
| 3D Unet | 4.00 | 2.67 | 3.80 | 3.56 | 3.51 | 6.47 |
| SCN | 3.71 | 2.43 | 3.67 | 3.43 | 3.75 | 5.26 |
| DRM | 2.72 | **1.60** | 2.00 | **1.97** | 1.94 | 6.09 |
| SA-LSTM | 2.44 | 1.95 | **1.88** | 2.15 | **1.79** | 4.45 |
| **Proposed** | **2.13** | 1.75 | 1.99 | 2.08 | 2.01 | **2.81** |

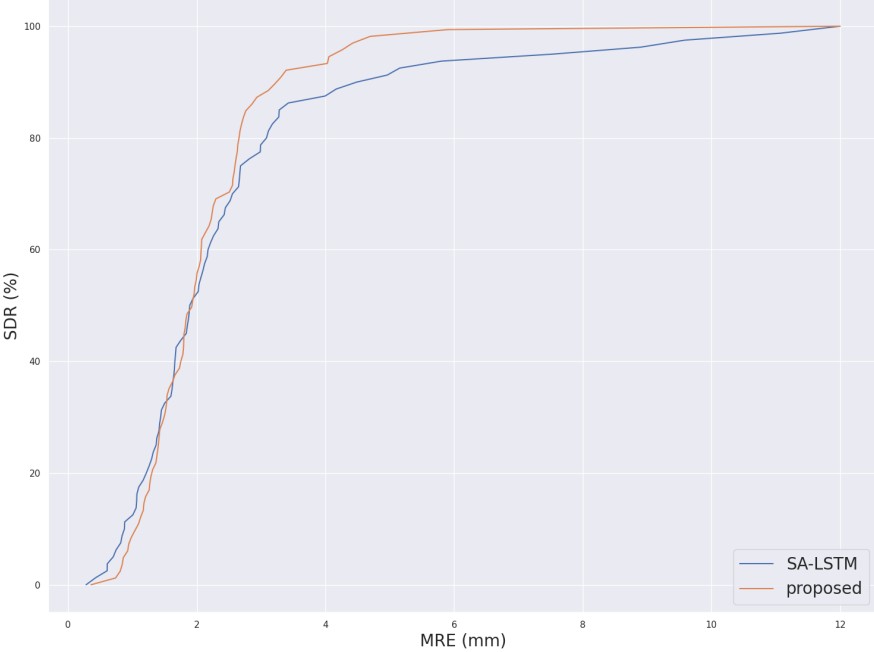

Figure A.1: SDR curves from the proposed method and SA-LSTM under different radii.

**A.2 More results on the in-house prostate dataset**

Table A.2 shows the MRE for each individual landmark of the in-house prostate dataset, predicted by our proposed method and the three baseline methods. L_1 ∼ L_6 represent the endpoints of the maximum longitudinal diameter, the maximum transverse diameter as well as the maximum anteroposterior diameter. Results demonstrate that incorporating physicians' prior knowledge can enhance the localization accuracy of all landmarks.

Table A.2: Individual landmark localization performance comparisons on the in-house prostate dataset, in terms of MRE.

| Method | Overall | L_1 | L_2 | L_3 | L_4 | L_5 | L_6 |
|---|---|---|---|---|---|---|---|
| 3D Unet | 3.57 | 3.58 | 3.61 | 3.23 | 3.95 | **3.32** | 3.74 |
| SCN | 3.48 | 3.36 | 3.18 | 3.49 | 3.63 | 3.43 | 3.78 |
| DRM | 3.44 | 3.33 | 3.17 | 3.32 | 3.59 | 3.45 | 3.80 |
| **Proposed** | **3.29** | **3.16** | **3.04** | **3.02** | **3.52** | 3.43 | **3.61** |

**A.3 More results on the ablation study**

For the PDDCA dataset, physicians annotate the occipital bone landmark (Occ_bone) by referring to the Chin landmark. Further ablation analyses are conducted on these two landmarks. In Table A.3, we show that the three modules can all effectively improve the localization accuracy of ambiguous landmarks and successfully incorporate physicians' prior knowledge into our model.

Table A.3: Ablation analysis results on the two correlated landmarks of the PDDCA dataset.

| Landmark | Fusion | Axis attention | Slice detection | SDR (%) | | | |
|---|---|---|---|---|---|---|---|
| | | | | 2 mm | 3 mm | 4 mm | 8 mm |
| Chin | | | | 45.45 | **98.18** | **100.00** | **100.00** |
| | ✓ | | | 56.36 | 98.18 | 100.00 | 100.00 |
| | ✓ | ✓ | | 63.56 | 94.18 | 99.18 | 100.00 |
| | ✓ | ✓ | ✓ | **65.42** | 91.06 | 98.40 | 100.00 |
| Occ_bone | | | | 18.35 | 32.62 | 60.05 | 87.40 |
| | ✓ | | | 49.04 | 74.82 | 91.00 | 100.00 |
| | ✓ | ✓ | | 61.82 | 80.42 | 89.89 | 98.18 |
| | ✓ | ✓ | ✓ | **61.84** | **87.18** | **92.61** | **100.00** |

