# OpenReview forum: "Prior Guided 3D Medical Image Landmark Localization"
_MIDL.io/2023/Conference — MIDL 2023 Poster_

### Official Review · Reviewer_PGDW · 2023-02-03

**Confidence:** 4
**Preliminary Rating:** 3
**Recommendation:** Poster

**Summary:**

The paper tackles landmark localisation in the particular setting where annotations are correlated due to the human labelling, e.g. when measuring diameters of the prostate. The method is a rather convoluted combination of 3D-CNN regression, regional heatmap prediction and the newly proposed correlated (attention-based) detection. The method is evaluated on two datasets, one private/in-house and another public one and benchmarked against other single-step or multi-step approaches and yields good results.

**Strengths:**

- The task of landmark localisation is relevant and the addressed sub-topic (correlated landmarks) has not been much considered before
- Combining continuous regression with discretised heatmap prediction is interesting
- The authors compare against a reasonable number of related approaches on two datasets, hence the paper could also be seen as a small validation study.
- There is an ablation study on the use of fusion, attention and slice classification

**Weaknesses:**

- I am curious how the proposed method would compare against a two-step approach were the ellipsoidal shapes defined by the two perpendicular diameter lines are firstly used as a segmentation task. This would be a slightly different but more commonly researched way of addressing the inherent landmark correlation.
- A more direct comparison of the proposed continuous regression plus discretised heatmap prediction against the alternative coarse-to-fine approach from Chen TMI 2021 would be necessary
- The description of the method is not concise and it is very hard to understand e.g. how exactly the pooling-based attention (eq. 2-5) is generated. What is the definition of F_max, why does Eq. 4 not just yield a scalar per channel? Fig. 2 has no description or explanation in the caption.
- The approach can also be seen as somewhat incremental to Chen et al., which also included attention modules  in their coarse-to-fine pipeline
- It remains unclear how the step "decode the two different heatmaps, obtaining the locations of landmarks and the key slices" is done (unless the reader looks it up in the reference). I assume this is referred to as the "fusion" in Table 2, but attention is also defined as "fusion" so I do not know for sure.


**Deanonymize Review:**

yes

**Detailed Comments:**

In the current status of the paper I would not recommend acceptance. While there are certainly some merits the contribution is rather limited and the method mainly optimises better towards annotation bias, which is interesting but does not directly imply the method itself is better for a general task

**Paper Type:**

both

**Questions To Address In The Rebuttal:**

- instead of showing a number of discrete success rates, a cumulative error plot would be preferable, since the precision of SA-LSTM on PDDCA is higher than "proposed", while the later improves robustness a statistical test would be required
- please give more details in Table 2 in the accompanying text what exactly is meant by the different steps.
- it would be informative to include visual results

Final justification:
While I am still somewhat unconvinced by the advantage of the proposed method compared to geometric (ellipsoid) fitting followed by a segmentation method in the instance of multiple annotated points on a 2D shape and the authors did not provide too much details about this additional experiment, I would give them the benefit of doubt. Furthermore I appreciate that they took great care addressing all points and clarifying some misunderstanding that led to weaknesses in the initial submission. Therefore I upgrade my score by 1.

---

### Official Review · Reviewer_BQgQ · 2023-02-06

**Confidence:** 3
**Preliminary Rating:** 3
**Recommendation:** Poster

**Summary:**

This work presents a novel two-stage coarse-to-fine method to localise landmarks in 3D medical images. Existing landmark localisation methods based on 3D convolutions, require downsampling of the image, compromising precise landmark localisation. Moreover, manually placed landmarks are often correlated to each other, e.g. in the same slice representing a diameter. This work addresses both problems by a coarse-to-fine approach, as well as a pipeline to merge features of correlated landmarks. Based on experiments on two very different datasets, this method out-performs existing single-stage methods, and performs on-par/outperforms existing coarse-to-fine methods.

**Strengths:**

- The novel contributions of the paper are well-articulated, and the paper is well-structured. The state-of-the-art methods and their main limitations are sufficiently addressed in the introduction.
- The quantitative comparison of the newly introduced method to multiple existing methods helps placing this work in perspective.
- I like the fact that two very different datasets were used to show that this method improves over existing methods. This displays the generalisability of the method.
- An ablation study was included, demonstrating the effectiveness of some newly introduced parts in the method.

**Weaknesses:**

- Although the main ideas of the paper are clear, there are some technical details in the proposed methodology that are not very clear. Figure 2 suggests that a single independent landmark is processed by the patch-based U-Nets multiple times, but from the text this is not apparent. Moreover, in section 2.2 a lot of processing of feature vectors is introduced. However, where each of these featuremaps ends up in the pipeline in Figure 2 is unclear to me.

- I think the authors could have addressed the differences between the independent and correlated features a bit more. The ablation study could have been extended by treating all the landmark points as independent, that would have fully proved the effectiveness of this approach. Now I am not 100% convinced. Moreover, it is unclear when exactly landmarks classify as correlated.

- In the coarse stage, the authors use a ResNet architecture that performs coordinate regression. I wonder if a shift in the image data shifts the predicted landmark coordinates accordingly, and if a heatmap approach as done in the refinement stage would be beneficial to this extend.

**Deanonymize Review:**

no

**Detailed Comments:**

- Figure 2 could be improved in terms of clarity. First of all, in the coarse section $n$ landmarks are predicted, but from Fig. 2 it only looks like a single coordinate is predicted. Either making this visually clear or using vector notation here would resolve this. Moreover, in 2.2, many variables are introduced regarding the fine-scale part of the method for correlated landmarks. Adding these variables to Fig. 2 would make the pipeline design more clear.
- Two different datasets were described for performing experiments on: a public dataset containing head/neck CT's, and an in-house prostate dataset. From the prostate dataset it is clear that the diameter measurements are correlated with each other. In the PDDCA dataset, five landmarks were annotated by the authors (or a collaborating medical expert). Which of these landmarks are correlated with each other?
- In Section 2.2 the authors mention adding Gaussian noise to the predicted coordinates. Is this only added during training, to make the method more robust against small errors in the coarse stage, or also added at inference?
- The authors should give the downsampled resolution of the single-stage methods in the experiments section. This helps interpreting the much higher MRE values given in Table 1 for these methods.


**Paper Type:**

both

**Questions To Address In The Rebuttal:**

- The authors distinguish between independent and correlated landmarks. Both go through a different pipeline in the fine-scale stage. Is there a performance difference in detecting the independent and correlated landmarks? What happens if we process correlated landmarks the same as independent landmarks?
- What is the benefit of doing coordinate regression at the coarse stage over using the heatmap-based approach as used in the refinement stages? Does this regression compromise the translational equivariance of the first step of the method?

---

### Official Review · Reviewer_8TNH · 2023-02-07

**Confidence:** 4
**Preliminary Rating:** 3
**Recommendation:** Poster

**Summary:**

The proposed framework is a prior-guided coarse-to-fine landmark localization approach that combines the benefits of heatmap regression and coordinate regression and incorporates prior knowledge from physician annotations. The method consists of two stages: a coarse stage using coordinate regression and a fine stage using multiple Unet models for independent landmarks and a correlated landmark localization module for correlated landmarks. The approach has been evaluated on two datasets and showed superiority over other methods

**Strengths:**

The main contribution is the proposed correlated landmark regression module, which consists of a fusion module, an axis attention module, and a slice classification branch.

The experimental setup is appropriate and the dataset used for the evaluation is extensive. The results seem to outperform existing approaches.

The ablation experiments test the importance of the different components of the approach.

In general, the method for landmark identification is clearly explained. However, see detail comments for sentences that should be re-written.



**Weaknesses:**

The main contribution of the paper is limited as there are existing approaches for landmark identification that use heatmaps. Jin et al. (2021) proposed heatmap regression in a multi-scale fashion.

There are approaches for landmark identification that are not mentioned in the literature review. For example,

"Reinforcement Learning for Anatomical Landmark Detection in Chest Radiographs" by Wei Ke et al. (2019)
"Reinforcement Learning for Robust Landmark Detection in Medical Images" by Wei Ke et al. (2021)

Including priors in the analysis has been an extensive area of research. For example,  atlas-based methods, where a statistical model of the target landmark is generated from a large number of annotated images. Active Shape Models (ASMs), where a deformable shape model is initialized using prior knowledge of the target landmark's shape and then optimized to fit the observed image data. Active Appearance Models (AAMs), are a type of ASMs that model both the shape and texture of the target landmark.




**Deanonymize Review:**

no

**Detailed Comments:**

There are several sentences that should be re-written for clarity:

"Although most of the existing methods implicitly integrate structural knowledge, they are kind of weak that are usually inferred from the landmarks’ distributions."

"Compared with implicit structural knowledge, physicians’ prior knowledge is more direct and shall be more useful."

"Coordinate regression can well maintain global representation for the structure/shape, even if some local features are lost due to downsampling"

" It is noteworthy that cropping patches centering at each individual landmark predicted in the coarse stage shall reduce the diversity across training samples."

"we merge the outputted feature vectors via element-wise summation, producing an axis channel attention map"


**Paper Type:**

validation/application paper

**Questions To Address In The Rebuttal:**

1. Where reinforcement learning approaches considered? These approaches have high accuracy in the landmark identification task.
2. In Axis attention fusion, it is not clear how the patches are extracted, are they just neighboring patches in the image? How are the correlated patches determined?
3. In the Figure, it seems that the Unets from different landmarks are sharing information, is this the case?
4. If the landmarks are in different locations why would they be correlated in this stage?

---

### Meta-Review · Area_Chair_DAJ4 · 2023-02-23

**Recommendation:** Accept (Poster)
**Confidence:** 4

**Metareview:**

The paper proposes a coarse-to-fine method for landmark localization in two datasets. This is a problem that has been extensively studied. The novelty of the current work lies in considering points as correlated or interdependent instead of independent. The authors exploit this for better results.

All three reviewers consider this paper borderline, with which I concur. The paper is not always easy to read, and the method description could be improved following the reviewer comments. Moreover, the paper contains quite some typos and sloppy punctuation. However, most of these could be easily corrected using a tool like Grammarly, and I strongly advice the authors to do so before the camera-ready version. Strengths are the evaluation in two very different datasets, a comparison to other landmark detection methods, and an ablation study. Considering landmarks as not just independent points but correlated points might find more applications, and the authors have demonstrated that this works. Hence, I am leaning toward acceptance.